# ROS-Mediated Necroptosis Promotes Coxsackievirus B3 Replication and Myocardial Injury

**DOI:** 10.3390/microorganisms13102389

**Published:** 2025-10-17

**Authors:** Junbo Huang, Yanjun Di, Qing Song, Zhiyun Cheng, Hao Wu, Mei Wu, Minjian He, Genrui Zhang, Fucai Wang, Lei Tong

**Affiliations:** School of Medicine, Huaqiao University, Quanzhou 362021, China; 17369909003@163.com (J.H.); 2126203012@stu.hqu.edu.cn (Y.D.); 19607367883@163.com (Q.S.); chengzhiyun@hqu.edu.cn (Z.C.); 2134133009@stu.hqu.edu.cn (H.W.); wumei200212@163.com (M.W.); 2134112003@stu.hqu.edu.cn (M.H.); 2234131030@stu.hqu.edu.cn (G.Z.); wfc@hqu.edu.cn (F.W.)

**Keywords:** coxsackievirus B3, viral myocarditis, necroptosis, RIP1, RIP3, ROS

## Abstract

Coxsackievirus B3 (CVB3) is a primary causative agent of viral myocarditis (VMC), which can lead to both acute and chronic cardiac inflammation accompanied by progressive heart failure and arrhythmias. Although CVB3 has been implicated in various forms of programmed cell death, whether it triggers necroptosis and the underlying mechanisms remains unclear. This study aimed to investigate the role and mechanism of CVB3-induced necroptosis and its effect on viral replication. Using both in vitro and in vivo models, we demonstrated that CVB3 infection significantly upregulates the expression of key necroptotic markers RIP1 and RIP3 in HeLa cells and mouse myocardial tissues. This upregulation was accompanied by elevated intracellular reactive oxygen species (ROS) levels and suppression of the Nrf2/HO-1 antioxidant pathway. Intervention with the necroptosis inhibitor Necrostatin-1 (Nec-1) or the ROS scavenger N-acetylcysteine (NAC) markedly attenuated cell death, suppressed viral replication, and ameliorated myocardial injury and inflammatory responses in infected mice. Mechanistically, CVB3 inhibits the Nrf2/HO-1 pathway, thereby inducing substantial ROS accumulation that promotes necroptosis. This effect can be reversed by NAC treatment. Our study reveals a novel mechanism through which CVB3 induces ROS-dependent necroptosis via the suppression of the Nrf2/HO-1 pathway, providing new insights into the pathogenesis of viral myocarditis and suggesting potential therapeutic strategies.

## 1. Introduction

VMC is an inflammatory condition of the heart muscle, frequently triggered by viral infections, particularly those from the enterovirus family, such as CVB3 [1,2]. CVB3 is recognized as a primary etiological agent of viral myocarditis, leading to significant cardiac dysfunction and, in severe cases, dilated cardiomyopathy [3]. CVB3 primarily targets cardiomyocytes, initiating inflammation, cell death, and fibrosis that disrupt normal heart function [4]. Upon infection, CVB3 triggers an acute inflammatory response characterized by the infiltration of immune cells, including macrophages and T-lymphocytes, which release cytokines and chemokines that exacerbate cardiac damage [5]. CVB3 and other cardiotropic viruses have been shown to activate pivotal inflammatory pathways, such as NF-κB and MAPK signaling cascades [6,7,8]. This activation stimulates the release of pro-inflammatory cytokines, ultimately leading to myocardial injury and driving the progression of viral myocarditis.

Research on CVB3-induced cell death has encompassed multiple forms, such as apoptosis, autophagy, pyroptosis, ferroptosis, and necroptosis [9,10,11,12,13]. The mechanisms by which CVB3 induces myocarditis and influences viral replication are not yet fully understood. Necroptosis is a form of programmed cell death characterized by cellular swelling, membrane rupture, and the release of intracellular contents, which can trigger an inflammation response. The process of necroptosis is initiated by the activation of tumor necrosis factor receptor 1 (TNFR1) through a cascade of signal molecules, primarily receptor-interacting protein kinases 1 and 3 (RIP1 and RIP3) [14,15]. Upon stimulation, these kinases form a complex known as the necrosome, which subsequently activates mixed lineage kinase domain-like protein (MLKL). The translocation of MLKL to cellular membranes results in osmotic swelling and rupture, leading to necrotic cell death [16]. Necroptosis is alleviated in the presence of the RIP1 inhibitor or RIP3 inhibitor [17,18].

Recent evidence suggests a critical link between ROS, nuclear factor erythroid 2-related factor 2 (Nrf2), and necroptosis in the context of viral infection. ROS, mainly generated during mitochondrial oxidative phosphorylation [19], can disrupt cellular function and trigger oxidative stress. When antioxidant defenses are overwhelmed, high levels of ROS can initiate cell death [20]. Notably, ROS plays a pivotal role in promoting necroptosis by modulating its key signaling pathways. For instance, respiratory syncytial virus (RSV) infection induces RIP3-MLKL-dependent necroptosis in macrophages, a process that is reversible by reducing ROS production [21]. Furthermore, ROS scavengers like NAC have been shown to inhibit particle-induced necroptosis [22], and studies indicate that ROS and necroptosis can engage in a positive feedback loop [23,24,25]. The transcription factor Nrf2 counteracts this oxidative stress by activating the expression of antioxidant enzymes such as heme oxygenase-1 (HO-1) [26]. The Nrf2/HO-1 pathway thus serves as a crucial cellular defense mechanism, scavenging excessive ROS and potentially mitigating ROS-induced necroptotic cell death.

In this study, we demonstrate that CVB3 infection induces necroptosis, which in turn promotes viral replication. We further show that CVB3 induces necroptosis by increasing ROS levels through the inhibition of Nrf2 and HO-1 expression.

## 2. Materials and Methods

### 2.1. Mice

Male Balb/c mice (5 weeks old) were purchased from the Wu’s Experimental Animals (Fuzhou, China). Mice were housed in a room with a constant temperature and humidity control, maintained under a 12/12 h light/dark cycle, and had free access to food and water. All experimental procedures were approved by the Ethics Committee of Huaqiao University (approval number: A2024071). Mice were inoculated with CVB3 through intraperitoneal injection to establish a viral myocarditis model [27]. Starting at 6 h post-infection, Nec-1 (HY-15760, MCE, Shanghai, China) was administered intraperitoneally daily at a dose of 0.8 mg/kg. Control mice received the same dose of Nec-1. Body weight changes were monitored daily. Mouse ventricles were used for histological examination and RNA/protein extraction.

### 2.2. Cell and Virus

HeLa cells (T8969, Abmgood, Shanghai, China) were cultured in Hela Cell Complete Medium (CM-0101, Procell, Wuhan, China) with 5% CO_2_ at 37 °C. Virus-infected cells were maintained in Hela Cell Complete Medium. CVB3 Woodruff (GenBank: U57056.1) and EGFP-CVB3 [28] were kindly provided by Prof. Zhaohua Zhong (Department of Microbiology, Harbin Medical University, Harbin, China). Viruses were propagated in HeLa cells and stored at −80 °C. Cells were infected with the virus in serum-free medium for 2 h, and cell lysates were collected and quantified using the conventional TCID50 assay.

### 2.3. TCID50 Assay

HeLa cells in the logarithmic growth phase were seeded into 96-well culture plates (FUV961, Beyotime, Shanghai, China). Upon reaching > 85% monolayer confluence, CVB3 solutions were serially diluted 10-fold from (10^−1^ to 10^−10^). For each dilution, 100 µL was added to eight duplicate wells followed by a 2-h infection period. After removing the viral supernatant, fresh complete culture medium was added, and the cells were incubated at 37 °C with 5% CO_2_ for 5–7 days, with daily observation for cytopathic effect (CPE). The 50% tissue culture infective dose (TCID50) was calculated using the Reed–Muench method.

### 2.4. Drug Treatment

NAC (HY-134495, MCE, Shanghai, China) was dissolved in PBS (C0221A, Beyotime, Shanghai). Nec-1 was dissolved in DMSO (ST038, Beyotime). Cells were pretreated with medium containing Nec-1 or NAC for 2 h, after which the supernatant was removed and cells were infected with CVB3 for 2 h. Following infection, cells were cultured in the medium containing Nec-1 or NAC for the indicated times.

### 2.5. Blood Tests

Following blood collection from the ocular vein, whole blood samples were allowed to stand at room temperature for 30 min and then centrifuged at 3000 rpm for 15 min to obtain serum. The concentrations of lactate dehydrogenase (LDH) and creatine kinase isoenzyme MB (CK-MB) in the serum were determined using a Hitachi automatic biochemical analyzer (Hitachi, Shanghai, China), via the rate method and immuno-inhibition method, respectively. Each sample was tested in triplicate, and the final concentrations were expressed in units per liter (U/L) and calibrated against a standard curve.

### 2.6. Western Blot

Protein was extracted from cells or tissues with RIPA lysate (P0013B, Beyotime) containing protease inhibitor PMSF (100:1 *v*/*v*) (P1045, Beyotime). After quantification with the BCA Protein Assay Kit (P0010S, Beyotime), sample buffer (P0015L, Beyotime) was added to the proteins, which were then separated by SDS-PAGE and transferred to the PVDF membrane for further analysis. The protein content in each lane was consistent. The proteins were blocked with 5% skimmed milk (P0216, Beyotime) at room temperature, and incubated with the appropriate primary antibody overnight at 4 °C. Blots were further incubated at room temperature with secondary antibody for 1 h. Blots were detected by BeyoECL Plus (P0018S, Beyotime), and analyzed as gray values using ImageJ software (version 1.53k; National Institutes of Health, Bethesda, MD, USA). Details of the antibodies used in this study, including their dilution ratios and sources, can be found in Appendix A. Anti-CVB3 3D was a gift from Prof. Zhaohua Zhong (Department of Microbiology, Harbin Medical University).

### 2.7. RT-qPCR

Total RNA was isolated using the RNA-Quick Purification Kit (RN001, ESScience, Shanghai, China). Quantitative PCR was performed using the Hiscript QRT SuperMix (RA103, Vazyme, Nanjing, China) and the Taq Pro Universal SYBR qPCR Master Mix (Q712, Vazyme). Gene expression was analyzed using the 2^−△△Ct^ method normalized to GAPDH. Primers were synthesized by Sangon (Shanghai, China). The primer sequences can be found in Appendix A.

### 2.8. Immunofluorescence Staining

HeLa cells were fixed in 4% formaldehyde (P0099, Beyotime) for 30 min and washed three times with PBS. Following fixation, the cells were permeabilized using 0.1% Triton X-100 (P0096, Beyotime) for 30 min at room temperature, then blocked with 1% BSA (ST025, Beyotime). After removing the blocking solution, the cells were incubated overnight at 4 °C with anti-RIP1 antibody and anti-RIP3 antibody, followed by a 30-min incubation with CY3-conjugated anti-rabbit IgG (A0516, Beyotime) or FITC-conjugated anti-rabbit IgG (A0562, Beyotime). The nuclei were stained with DAPI for 5 min, rinsed with PBS and then sealed with the fluorescence quencher (P0122, Beyotime). Images were captured under the fluorescence microscope (Nikon, Shanghai, China).

### 2.9. Histopathology

Mouse heart tissues were obtained, and the ventricles were dissected, fixed in 4% formaldehyde, and embedded in paraffin (C0175A, Beyotime). Cardiac tissues were sectioned into 5 μm thick slices, deparaffinized with xylene (1330-20-7, J&K, Shanghai, China), rehydrated through a gradient ethanol (R003459, Rhawn, Shanghai, China) series (100%, 95%, 80%, 70%), stained with hematoxylin and eosin (H&E) (C0105S, Beyotime), and observed for inflammatory and pathological changes under the light microscope (Nikon, Shanghai, China).

### 2.10. Cell Counting

Cells were stained according to the instructions of the Staining Kit (C0040, Solarbio, Beijing, China), and viable cells were counted using a hemocytometer (Nikon, Shanghai). Live cells appear transparent or colorless after staining with Trypan Blue (ST798, Beyotime), while dead cells appear blue. Relative PI-positive cells were counted in the CVB3 group or CVB3+Nec-1 group, normalized to total cell count, and presented relative to the mock group.

### 2.11. PI Staining

After gently removing the medium from the wells, 10 µM of Propidium iodide (PI) (MB2920, Meilunbio, Dalian, China) working solution was added, and the cells were incubated for 30 min. The morphological changes were observed and images were captured using fluorescence microscopy.

### 2.12. Detection of ROS

HeLa cells were pretreated with NAC for 1 h and infected with CVB3 for 24 h. Cells were stained with 10 µM 2′,7′-dichlorofluorescein diacetate (DCFH-DA) (D103583, Aladdin, Shanghai, China) at 37 °C for 30 min protected from light. Cells were washed twice with pre-cooled PBS and analyzed by fluorescence microscopy at an excitation wavelength of 485 nm and an emission wavelength of 520 nm. Fluorescence intensity was analyzed using ImageJ (version 1.53k; National Institutes of Health, Bethesda, MD, USA).

### 2.13. Statistical Analyses

GraphPad Prism 9.0 software (version 9.0.0; GraphPad Software, San Diego, CA, USA) was utilized for statistical analysis. The data were presented as mean ± standard deviation (SD). Student’s *t*-test was used to analyze the results. A *p* value < 0.05 was considered as statistically significant. Three biological replications were performed for each experimental group.

## 3. Results

### 3.1. CVB3 Infection Induces Necroptosis in HeLa Cells

CVB3, a major causative agent of VMC, induced significant cytopathic effects in HeLa cells [29]. At 24 h post-infection (hpi), infected cells exhibited marked morphological swelling as observed by light microscopy (LM), with a substantial proportion detaching and forming clustered aggregates. To assess membrane integrity, we performed PI staining (Figure 1A). Quantitative analysis revealed that the number of PI-positive cells following CVB3 infection was significantly increased, approximately 5.0-fold higher than that in the control group (Figure 1B). PI staining revealed membrane rupture in CVB3-infected cells, a hallmark of necroptosis, indicating that CVB3 triggers this form of cell death. Together, these findings suggest that CVB3 triggers necroptosis-mediated cell death.

To further confirm CVB3-induced necroptosis, HeLa cells were infected with CVB3 (MOI = 5), then collected at 0, 3, 6, and 9 hpi, and subjected to Western blot analysis. As shown in Figure 1C,E, protein levels of RIP1, RIP3, and P-MLKL were significantly upregulated at 6 and 9 hpi. Moreover, Figure 1D,F demonstrated that the expression of these necroptosis-related proteins correlated with increasing viral titers. The temporal activation of necroptotic signaling paralleled viral replication, suggesting a direct mechanistic relationship. Taken together, these results demonstrate that CVB3 infection induces necroptosis in HeLa cells.

### 3.2. CVB3 Induces Necroptosis via the RIP1/RIP3 Pathway

To further elucidate the mechanism of necroptosis induced by CVB3, we investigated the effect of the RIP1 inhibitor Nec-1. HeLa cells were pretreated with Nec-1 for 2 h, infected with CVB3 (MOI = 1) for 2 h, and then maintained in medium containing Nec-1 for 24 h (Figure 2A). Results showed that CVB3-induced cell death was significantly attenuated by Nec-1 treatment, as assessed by PI staining (Figure 2B,C). Western blot analysis showed that Nec-1 treatment significantly reduced CVB3-mediated upregulation of both RIP1 and RIP3 (Figure 2D,E). This observation was further confirmed by immunofluorescence staining, where CVB3-induced increases in RIP1 and RIP3 expression were markedly suppressed by Nec-1 treatment (Figure 2F). These results demonstrate that CVB3 triggers necroptosis in HeLa cells through activation of the RIP1/RIP3 signaling pathway.

### 3.3. Necroptosis Promotes CVB3 Replication In Vitro

To evaluate whether the CVB3-induced necroptosis affects viral production, we examined the impact of Nec-1 on CVB3 genome replication and CVB3 3D expression via RT-qPCR and Western blotting at 24 hpi. The results confirmed that Nec-1 significantly reduced CVB3 genomic RNA levels (Figure 3A). Western blotting further demonstrated a marked decrease in CVB3 3D expression in Nec-1-treated cells (Figure 3C,D). TCID50 assays showed that Nec-1 suppressed progeny virus production by approximately 40% (Figure 3B). The EGFP-CVB3 variant (MOI = 0.5) was used to infect HeLa cells after treatment with Nec-1. The EGFP expression was observed at 24 hpi by fluorescence microscopy. Microscopic observation showed that fewer progeny viruses were produced in the infected cells pretreated with Nec-1 (Figure 3E), further supporting the inhibitory effect of necroptosis blockade on viral replication. Collectively, these results indicate that CVB3-induced necroptosis enhances viral genome replication and progeny production.

### 3.4. Inhibition of CVB3-Induced Necroptosis Reduces Viral Replication and Ameliorates Myocardial Injury In Vivo

To evaluate the role of necroptosis in CVB3-induced VMC, CVB3-infected 5-week-old Balb/c mice were intraperitoneally inoculated with 10^6^ TCID50 of CVB3. The experimental group received daily 0.8 mg/kg Nec-1 (i.p.) for 7 consecutive days starting 12 h post-inoculation, while controls received an equivalent volume of PBS. Cardiac tissues and serum harvested at 5 and 7 days post-infection (dpi) were systematically subjected to histopathological analysis (Figure 4A). CVB3-infected mice showed a significant decrease in body weight by day 4 post-infection. By day 7, the body weight in CVB3-infected mice declined by approximately 19.6% compared to control group. In contrast, infected mice treated with Nec-1 showed significant relief in weight loss on the 7th day after infection (Figure 4B). Histopathological examination revealed severe myocarditis in CVB3-infected mice, characterized by cardiomyocyte necrosis and inflammatory cell infiltration. Nec-1 treatment attenuated these pathological changes, with reduced monocyte infiltration and preserved tissue architecture, suggesting that Nec-1 ameliorated myocardial damage via necroptosis inhibition (Figure 4C). Additionally, serum levels of myocardial injury markers (CK-MB and LDH) were significantly lower in Nec-1-treated mice, decreasing by 40.2% and 49.9%, respectively (Figure 4D). RT-qPCR analysis showed that CVB3-induced pro-inflammatory cytokines (TNF-α, IL-6, IL-1β, and IFN-α) were significantly decreased in Nec-1-treated myocardium (Figure 4E).

To assess viral replication, total RNA and proteins were extracted from the myocardium. CVB3 RNA and 3D protein were determined by RT-qPCR and Western blotting, respectively. Both CVB3 RNA abundance and 3D protein levels were significantly reduced in Nec-1-treated myocardium, indicating inhibited viral replication (Figure 4F,G). To elucidate the mechanism of Nec-1 in VMC mice, the expression levels of RIP1 and RIP3 were determined by Western blotting. Nec-1 treatment significantly inhibited RIP1 and RIP3 levels in CVB3-infected cardiac tissue (Figure 4G,H). These results demonstrate that necroptosis inhibition via the RIP1/RIP3 pathway not only restricts CVB3 replication but also mitigates myocardial damage and inflammation.

### 3.5. CVB3 Induces Necroptosis via ROS Production in HeLa Cells

Previous studies have shown that necroptosis is associated with elevated levels of ROS [30], and CVB3 replication in cardiomyocytes is correlated with the production of ROS [31]. To investigate the relationship between ROS and CVB3-induced necroptosis, we detected oxidative stress indicator proteins. HO-1, a downstream target of Nrf2, plays a key role in protecting against oxidative stress-mediated damage. Given the established association between necroptosis and oxidative stress, we investigated whether CVB3-induced necroptosis is mediated by ROS. The expression of Nrf2 and HO-1 was downregulated in murine cardiac tissues following CVB3 infection. Western blot analysis confirmed a significant reduction in their levels at 5 days post-infection (dpi), followed by a further decline at 7 dpi. (Figure 5A,B). Consistent with these in vivo findings, CVB3-infected HeLa cells (MOI = 1) demonstrated time-dependent decreases in Nrf2 and HO-1 at 12, 24, and 36 h post-infection (Figure 5C,D). Concurrently, CVB3 infection elevated intracellular ROS levels, as measured by DCFH-DA fluorescence. To confirm ROS involvement, we treated cells with the ROS scavenger NAC, which effectively reduced CVB3-induced ROS accumulation (Figure 5E). NAC treatment also inhibited CVB3-induced necroptotic cell death, as assessed by PI staining and cell viability assays (Figure 5F). Western blot analysis showed that increasing doses of NAC remarkably decreased the expression of RIP1, RIP3, and the viral 3D protein (Figure 5G,H). Meanwhile, NAC treatment inhibited the accumulation of viral genomic RNA and the production of progeny virus particles, as measured by RT-qPCR (Figure 5I) and TCID50 assay (Figure 5J), respectively. These findings indicate that CVB3 infection disrupts the Nrf2/HO-1 antioxidant pathway, leading to ROS accumulation, which activates RIP1/RIP3-mediated necroptosis and enhances viral replication.

## 4. Discussion

Myocarditis is an inflammatory heart disease with diverse etiologies, among which CVB3-induced viral myocarditis is the most prevalent form. Although supportive care can temporarily alleviate heart failure symptoms in myocarditis patients, no antiviral therapies or effective vaccines against CVB3 have been developed to date [32,33]. Therefore, further investigation into the molecular mechanisms of CVB3-induced viral myocarditis is essential for developing effective treatments. Recent studies on CVB3-induced cell death have primarily focused on apoptosis and autophagy, whereas the relationship between CVB3 and necroptosis remains relatively scarce. Necroptosis is a form of programmed cell death mediated by specific molecules. Multiple viruses such as Vaccinia virus (VV) [34], Cytomegalovirus (CMV) [35,36], Herpes simplex virus (HSV) [37], Influenza A virus (IAV) [38], and Severe Acute Respiratory Syndrome Coronavirus 2 (SARS-CoV-2) [39] have been shown to either activate or inhibit this process. The molecular mechanisms of necroptosis involve sequential activation of key signaling molecules. Upon stimulation by death signals, RIP1 and RIP3 interact to form a necroptosome, which subsequently activates downstream MLKL. This activation promotes cell membrane rupture and the release of cellular contents [40]. This process disrupts cellular homeostasis and releases damage-associated molecular patterns into the extracellular space, thereby activating immune cells and triggering inflammatory responses. A deeper understanding of this mechanism may elucidate the connection between cardiomyocyte death and CVB3-induced viral myocarditis. Therefore, we investigated the molecular mechanisms of CVB3-induced necroptosis and proposed novel therapeutic strategies for viral myocarditis. PI staining revealed the loss of cell membrane integrity in CVB3-infected HeLa cells, indicating membrane rupture. This rupture is a hallmark of necroptosis, a form of regulated cell death mediated by RIP1/RIP3/MLKL activation. Our data demonstrated that CVB3 infection upregulated RIP1, RIP3, and P-MLKL in HeLa cells, consistent with necroptosis activation. Nec-1, a RIP1 inhibitor, effectively blocks necroptosis. We found that Nec-1 inhibited CVB3-induced cell death and RIP1/RIP3 upregulation in vitro and in vivo. This aligns with prior studies showed that necrosome components like RIP1/RIP3 were essential for viral-induced necroptosis. Our findings on CVB3-induced necroptosis are supported by previous reports of virus-triggered necroptosis, including IAV-ZBP1-RIP3-dependent necroptosis [41] and Reovirus-induced RIP1-dependent necroptosis in L929 cells [42], both similar to our findings in CVB3 infection.

A paradoxical role of virus-induced necroptosis has been reported, whereby this programmed cell death pathway may either promote or inhibit viral replication. For instance, murine cytomegalovirus induces necroptosis in immune cells such as macrophages, thereby restricting viral dissemination and replication [35]. Nogusa et al. demonstrated that RIP3-activated MLKL-driven necroptosis confers protective immunity in mouse embryonic fibroblasts against IAV [43]. In contrast, Zhang et al. showed that necroptosis enhances Coxsackievirus A6 production in human rhabdomyosarcoma cells [44]. These findings suggest that viruses can exploit host cell necroptosis to facilitate their replication or evade immune surveillance, prompting the hypothesis that necroptosis may facilitate CVB3 replication. In this study, we demonstrate that necroptosis enhances CVB3 replication both in vitro and in vivo, as validated by measuring CVB3 RNA, CVB3 3D protein levels, and TCID50. Collectively, these results suggest that necroptosis facilitates CVB3 replication, potentially through the inflammatory microenvironment induced by viral infection. However, further analyses are necessary to elucidate the precise mechanisms involved.

Our in vivo experiments demonstrated that pharmacological inhibition of necroptosis using Nec-1 significantly attenuated the pathological damage to the VMC heart tissues, ameliorated weight loss, and suppressed the elevation of myocardial injury markers (CK-MB and LDH), which may be related to the downregulation of RIP3 and RIP1 expression. The therapeutic effects of Nec-1 on viral myocarditis in mice are consistent with previous findings [13,45]. Collectively, our results suggested that necroptosis contributes substantially to CVB3-induced myocarditis by promoting cardiomyocyte death and triggering inflammatory responses. These data highlighted the necroptosis pathway as a promising therapeutic strategy for viral myocarditis induced by CVB3 infection.

ROS play a critical role in inflammation as signaling molecules that activate immune responses. However, excessive ROS can drive chronic inflammation by activating nuclear factor kappa B, leading to persistent tissue damage [46,47,48]. Studies have demonstrated that mito-TEMPO, a mitochondrial superoxide scavenger, protects the heart by inhibiting CVB3-induced myocardial ROS elevation and suppressing NLRP3 inflammasome activation [49]. Notably, ROS mediates not only inflammation but also necroptosis. Studies have shown that ROS oxidize specific cysteine residues on RIPK1 and RIPK3, an oxidative modification that alters their conformation and promotes the formation of the RIPK1-RIPK3 necrosome complex, a crucial step in necroptosis initiation [50]. Fan et al. reported that 1,3-Dichloro-2-propanol induces necroptosis in rat renal cells via ROS-dependent pathways [51]. Tian et al. demonstrated that the ROS scavenger NAC reduces iron overload induced ROS elevation, thereby alleviating necroptosis in osteoblasts [52]. Huang et al. discovered that upon inflammasome activation in macrophages, mitochondrial ROS (mtROS) promoted the binding of Gasdermin-D to the mitochondrial membrane, releasing mtROS and facilitating RIP1/RIP3/MLKL-dependent necroptosis [53]. To investigate whether ROS contribute to CVB3-induced necroptosis, we evaluated the effects of NAC in HeLa cells. Our results indicated that ROS was significantly reduced, as evidenced by decreased PI staining, improved cell viability, and downregulation of RIP1 and RIP3 expression. Notably, NAC also suppressed CVB3 replication, suggesting a potential link between ROS-mediated necroptosis and viral propagation. These results indicated that ROS was critical for mediating CVB3-induced necroptosis. In conclusion, as key facilitators of inflammation and necroptosis, ROS underscore the necessity for further research into therapeutic strategies that regulate ROS levels to alleviate inflammatory responses and improve disease outcomes. To further explore the mechanisms by which CVB3 induced ROS production in cells, Mechanistically, CVB3 infection downregulated the antioxidant proteins Nrf2 and HO-1 both in vivo and in vitro, consistent with prior reports. These findings highlight ROS as a key mediator of CVB3-induced inflammation and necroptosis, suggesting that the Nrf2/HO-1 pathway may be a potential therapeutic target for viral myocarditis. The mechanism identified in HeLa cells require further confirmation in more heart-relevant models (such as primary cardiomyocytes or cardiac organoids) to fully elucidate their contribution to the pathophysiology of viral myocarditis.

Our study demonstrated that CVB3 infection induced necroptosis, which promoted viral replication both in vitro and in vivo. Pharmacological inhibition of necroptosis using Nec-1 significantly attenuated CVB3-induced myocardial injury and associated inflammatory responses. Mechanistically, we found that CVB3 may trigger necroptosis by suppressing the Nrf2/HO-1 antioxidant pathway, leading to ROS accumulation (Figure 6). These findings provided novel insights into the pathogenesis of CVB3-induced viral myocarditis and suggested that targeting the necroptosis pathway or modulating ROS levels through Nrf2/HO-1 activation could represent promising therapeutic strategies for this condition.

## Figures and Tables

**Figure 1 microorganisms-13-02389-f001:**
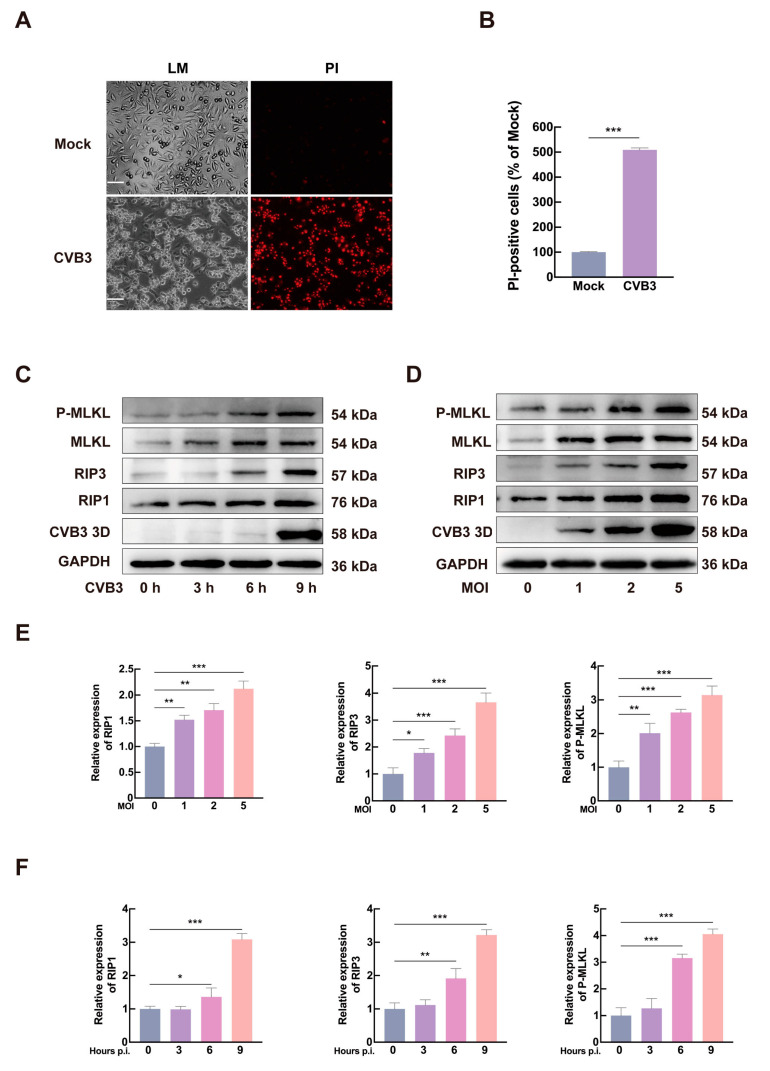
CVB3 induces necroptosis in HeLa cells. (**A**) PI staining (red) of HeLa cells at 24 h post-infection with CVB3 (MOI = 1) or mock treatment. Scale bar = 10 µm. (**B**) The relative count of PI-positive cells in the CVB3 group was calculated after normalization to total cell count and is presented relative to the mock-treated group. (**C**,**D**) Western blot analysis of RIP1, RIP3, and P-MLKL expression in CVB3-infected cells at indicated time points with viral titers measured in parallel. GAPDH was used as a loading control to normalize protein loading. (**E**,**F**) Bands in panels C and D were quantified by ImageJ, and the intensity ratios of RIP1, RIP3, and P-MLKL to corresponding GAPDH were shown. Data are represented as mean ± SD from three independent experiments. * *p* < 0.05, ** *p* < 0.01, *** *p* < 0.001.

**Figure 2 microorganisms-13-02389-f002:**
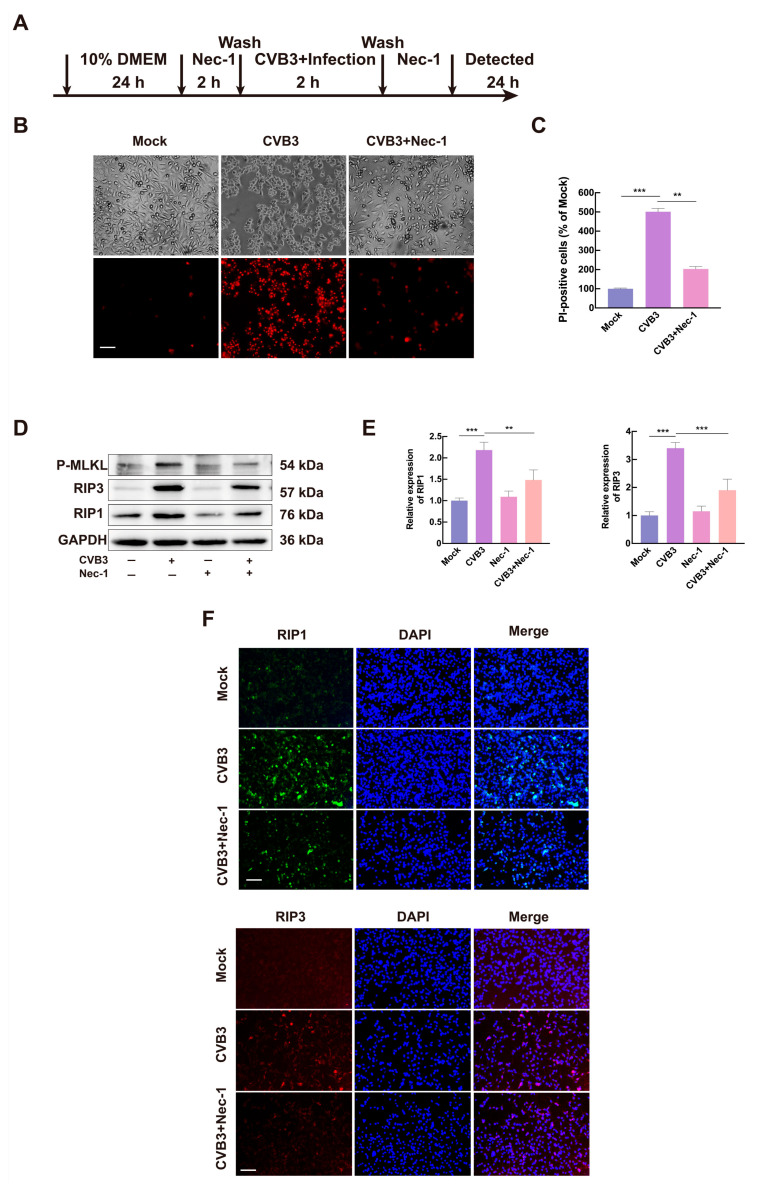
Nec-1 inhibits CVB3-induced necroptosis in HeLa cells. (**A**) Experimental timeline for Nec-1 treatment. HeLa cells were infected with CVB3 (MOI = 1) and treated with Nec-1 at indicated time points. The arrows indicated the start of each process. (**B**) HeLa cells infected with CVB3 (MOI = 1) were treated with Nec-1. At 24 hpi, cell morphology was analyzed by PI staining, with images captured using light and fluorescence microscopy. Scale bar = 10 µm. (**C**) Relative PI-positive cells in the CVB3+Nec-1 group, normalized to total cells and presented relative to the mock group. (**D**) HeLa cells were pretreated with 20 μM Nec-1 for 2 h before CVB3 infection (MOI = 1). Cells were harvested at 24 h post-infection, and lysates were analyzed by Western blot. (**E**) Quantitative analysis of RIP1 and RIP3 protein levels by ImageJ, normalized to GAPDH as a loading control. (**F**) RIP3 expression as assessed by fluorescence microscopy with Immunofluorescence staining of RIP1 and RIP3 in CVB3-infected cells (MOI = 1) at 24 hpi. Cells were stained with anti-RIP1 and anti-RIP3 antibodies, followed by FITC-conjugated (green) or CY3-conjugated (red) secondary antibodies, respectively. DNA was counterstained with DAPI (blue). Cell morphology was imaged by light microscopy, and merged images of RIP1/RIP3 with DAPI are shown. Scale bar = 10 µm. Data are represented as mean ± SD of three independent experiments. ** *p* < 0.01; *** *p* < 0.001.

**Figure 3 microorganisms-13-02389-f003:**
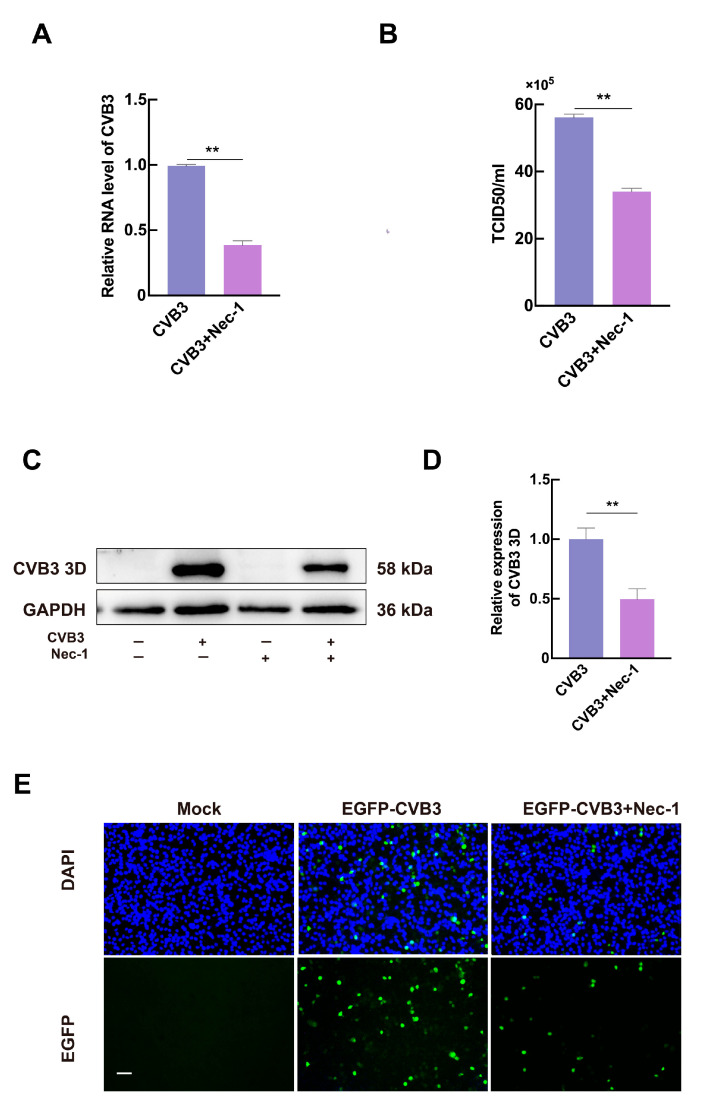
Nec-1 inhibits viral CVB3 production in HeLa Cells. (**A**) HeLa cells were infected with CVB3 at MOI of 1 and were cultured with Nec-1 for 24 h. Total RNA was extracted and viral CVB3 RNA was quantified by RT-qPCR. Total RNA was extracted and viral CVB3 RNA was quantified determined by RT-qPCR. (**B**) Total progeny viruses were collected in Nec-1-treated HeLa cells 24 h after infection with CVB3 (MOI = 1) using HeLa cells titration. (**C**,**D**) HeLa cells were preincubated with 20 μM Nec-1 for 2 h prior to CVB3 (MOI = 1) infection. Cellular lysates were analyzed by Western blot subjected to immunoblot analysis 24 h post-infection to determine 3D expression, with GAPDH utilized as the loading control. All data were normalized to GAPDH mRNA and are expressed as fold change relative to the CVB3 group, which was assigned a value of 1.0. (**E**) Fluorescence microscopy images of HeLa cells infected with EGFP-CVB3 (MOI = 5). Images were acquired 24 h post-infection. Scale bars = 10 µm. Data are represented as mean ± SD of three independent experiments. ** *p* < 0.01.

**Figure 4 microorganisms-13-02389-f004:**
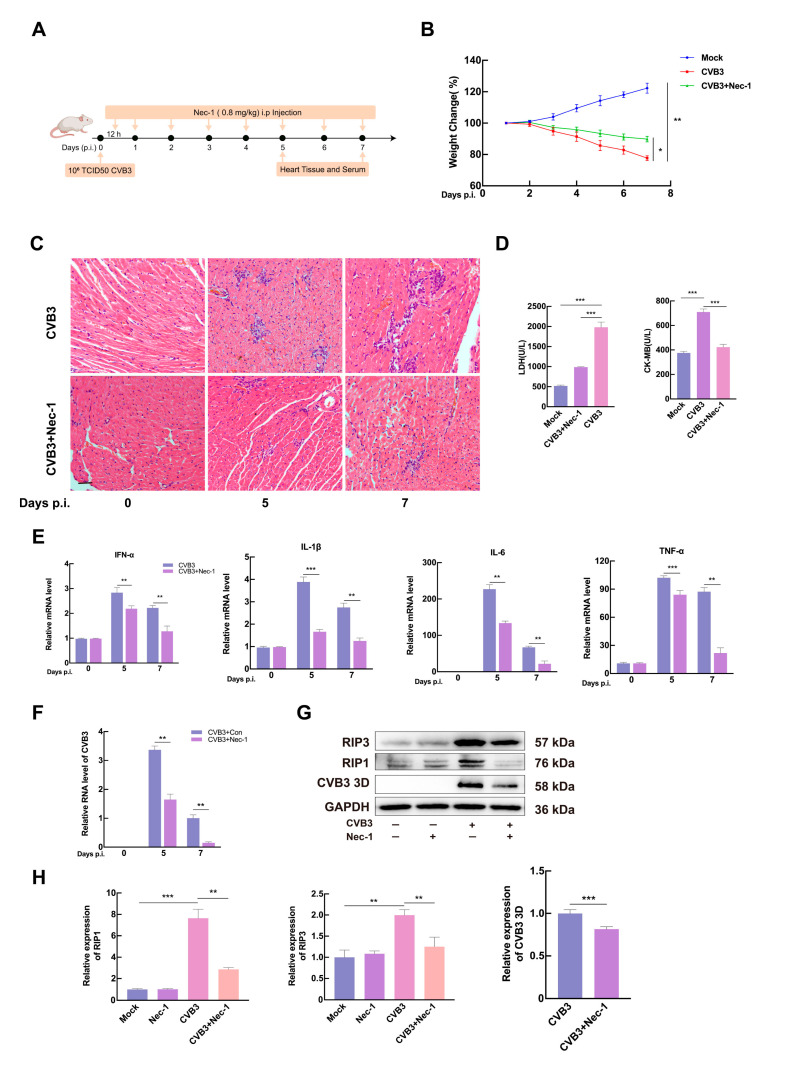
Nec-1 demonstrates cardio-protective effects in CVB3-infected mice. (**A**) Experimental design. (**B**) Daily body weight changes in mice. The arrows indicated the time points of virus infection, addition of Nec-1, and sample collection. (**C**) Histopathological assessment of cardiac lesions by H&E staining at 7 dpi in CVB3-infected mice with Nec-1 treatment. (**D**) Comparative analysis of serum LDH and CK-MB levels in Nec-1-treated versus untreated groups. (**E**,**F**) Myocardial inflammatory cytokines (IFN-α, IL-1β, IL-6, TNF-α) were quantified by RT-qPCR. Total RNA isolation from cardiac tissues was performed. Data are presented relative to the CVB3 group after normalization to GAPDH mRNA levels, with the control group value set to 1.0. (**G**,**H**) Cardiac RIP1, RIP3, and CVB3 3D protein expression at 7 dpi with/without Nec-1 treatment. (**G**) Representative immunoblots. (**H**) Quantified protein levels normalized to GAPDH. Data are represented as mean ± SD of three independent experiments. * *p* < 0.05; ** *p* < 0.01; *** *p* < 0.001.

**Figure 5 microorganisms-13-02389-f005:**
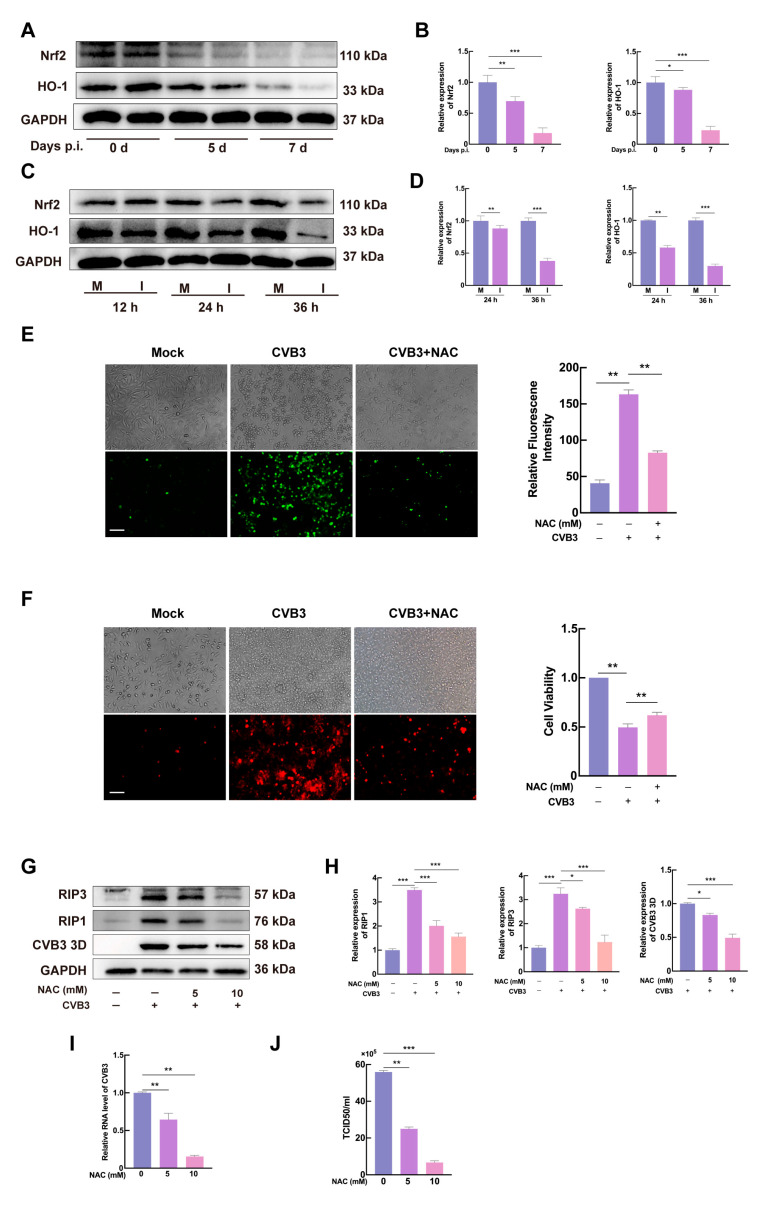
Excessive ROS induces necroptosis. (**A**,**B**) Myocardial tissues were, respectively, collected from control and CVB3-infected mice at 5 and 7 dpi. Nrf2 and HO-1 were analyzed by Western blot, using GAPDH as a loading control. (**C**) HeLa cells were mock-infected (M) or infected with CVB3 at a MOI of 1 (I) and then collected at the indicated times. Nrf2 and HO-1 were detected by Western blot analysis. GAPDH is shown as a loading control. (**D**) Densitometric quantification of Nrf2 and HO-1 to GAPDH bands was conducted using ImageJ with background subtraction. (**E**) Intracellular ROS production was detected by fluorescence microscopy with DCFH-DA staining, followed by quantitative analysis of signal intensity using ImageJ. Scale bar = 10 μm. (**F**) Cell membrane integrity assessment via PI staining in CVB3-infected HeLa cells (MOI = 1) cultured with NAC for 24 h. Phase-contrast and fluorescence imaging were simultaneously acquired. Cell viability was subsequently determined through CCK-8 absorbance measurements at 450 nm. Scale bar = 10 μm. (**G**) HeLa cells infected with CVB3 (MOI = 1) were treated with NAC (0, 5, 10 mM) for 24 h, and RIP1, RIP3 and CVB3 3D were detected by Western blotting. (**H**) Band intensities were quantified using ImageJ, followed by normalization to GAPDH expression levels for data standardization. (**I**) CVB3 RNA in NAC-treated HeLa cells at 24 hpi, quantified by RT-qPCR. Data are presented relative to the CVB3 group after normalization to GAPDH mRNA levels, with the control group value set to 1.0. (**J**) Progeny virus titers determined by TCID50 assay in NAC-treated (0, 5, 10 mM) and untreated cells at 24 h post-CVB3 infection (MOI = 1). Data are represented as mean ± SD of three independent experiments. * *p* < 0.05; ** *p* < 0.01; *** *p* < 0.001.

**Figure 6 microorganisms-13-02389-f006:**
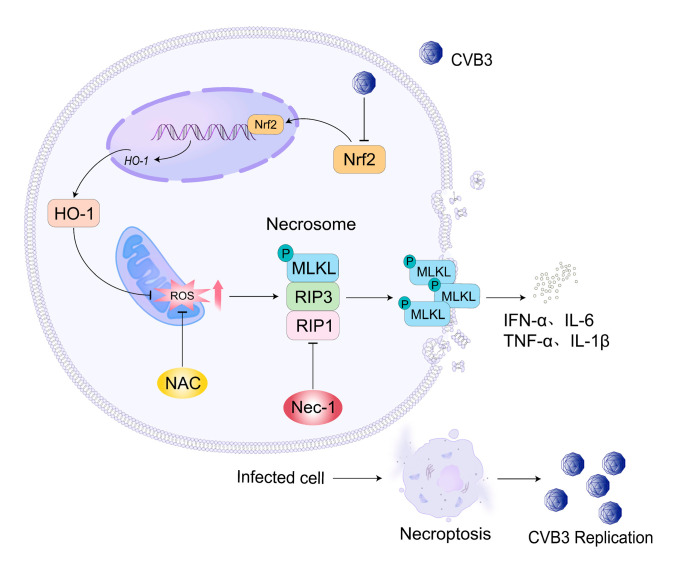
The mechanism of necroptosis by CVB3 infection.

## Data Availability

The data presented in this study are available on request from the corresponding author. The data are not publicly available due to privacy restrictions.

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
