# Peer review of "ROS-Mediated Necroptosis Promotes Coxsackievirus B3 Replication and Myocardial Injury"

_microorganisms, 2025, doi:10.3390/microorganisms13102389_

Round 1

Reviewer 1 Report

Comments and Suggestions for Authors

The results are interesting and suggested that targeting the necroptosis pathway or modulation ROS levels through Nrf2/HO-1 activation could represent promising therapeutic strategies for this condition. 

Author Response

In the article ROS-Mediated Necroptosis Promotes Coxsackievirus B3 Replication and Myocardial Injury authors presented the results of the study about CVB3 that may trigger necroptosis by suppressing the Nrf2/HO-1 antioxidant pathway, leading to ROS accumulation and these findings provided novel insights into the pathogenesis of CVB3-induced viral myocarditis.

The results are interesting and suggested that targeting the necroptosis pathway or modulating ROS levels through Nrf2/HO-1 activation could represent promising therapeutic strategies for this Condition.

Response: Thank you very much for taking the time to review this manuscript. 

Reviewer 2 Report

Comments and Suggestions for Authors

The authors provide a concise manuscript presenting data on CVB3 induced necroptosis in HeLa cells.

I have only some minor comments: 

1) In the introduction, please provide a brief overview on central inflammatory pathways induced by CVB3 and other viruses. 

2) Please check, whether you provide all company names or contact information (e.g. for ImageJ, Washington DC, etc)

3) Please include Figure 6 - as a central illustration more 

Reviewer 3 Report

Comments and Suggestions for Authors

The abstract does not mention the aim of the study or what you want to observe ? the manuscript directly told that CVB3 induces necroptosis in HeLa cells, as evidenced by loss of cell membrane integrity and increased expression of RIP1 and RIP3. This is not correct.

There is no explanation in the abstract about what remains unknown or which mechanisms need to be clarified.

The methods section in the abstract is too many explanation. Need to concise.

The results in the abstract are not clearly or adequately described.

The abstract needs to be completely rewritten.

The introduction should better explain the rationale and clearly state the knowledge gap this study aims to address.

The introduction would benefit from a more comprehensive review of previous studies investigating the relationship between the t CVB3 infection and necroptosis, which in turn promotes viral replication.

Do you have Ethics Committee of Huaqiao University approved number ? please show.

Mice were inoculated with CVB3 via intraperitoneal injection to establish a viral myocarditis model. What is your reference ? do you already established ?

Too many time repeat 37℃ with 5% CO2 (line 88 and 76).

Please explain more details Blood tests.

The following antibodies Line 110 is not sufficient. Please make it table where you can show dilution dactor, company name, secondary antibody name and company, time duration incubation.

Real-Time Quantitative PCR primer list should be in table and gene name.

Fig 1B incorrect quantification. You should calculate percentage as cell number may vary. 1A also not clear what you want to say as two different images. Parameter are not same.

Fig 1E and F relative expression level RIP1 or other do you compare with internal control ? why Control are not 1 (means 0h). Why not fold change ? Please learn how to do statistical analysis.

Line 215 what is (A)

Why you did not use Hoechst (for live cell) ? Fig 2B

Quantification wrong Fig 2C

I am not sure due to high number of cell, PI staining showed positive ? because I can clearly see the cell density too different.

Figure 6 too much general. Nothing mechanism.

I feel all data have some problem. This kind of data are not acceptable.

You should use Arial front inside of the images.

There are inappropriate uses of periods throughout the manuscript. Please review and revise punctuation.

Verb tenses should be consistent—ideally, use the past tense throughout.

The manuscript’s grammar requires improvement. Have you considered professional proofreading or English-language editing services? Please have the manuscript reviewed for grammatical accuracy by a native speaker.

Please learn how to show original blot. It should be in PPT with figure and marker lane. Not just submit the figure.

Reviewer 4 Report

Comments and Suggestions for Authors

The authors present a well-structured and timely study investigating the role of RIP1/RIP3-mediated necroptosis in Coxsackievirus B3 (CVB3) pathogenesis. The study is comprehensive, employing both in vitro (HeLa cells) and in vivo (mouse model) approaches to substantiate the claims. The data are generally clear and support the conclusions. However, some key issues require clarification.

1.Cell Line Justification: The use of HeLa cells (cervical adenocarcinoma) as the primary in vitro model for a study on viral myocarditis is a significant limitation. The authors should Include data from a cardiomyocyte model to confirm the key findings or explicitly discuss the limitation of using HeLa cells.

2.Mechanistic Link Between ROS and Necroptosis: The data show that CVB3 suppresses Nrf2/HO-1, increases ROS, and that scavenging ROS (with NAC) inhibits necroptosis and viral replication. However, the direct causal link between ROS and the activation of the necroptotic machinery (RIP1/RIP3/MLKL) is not fully established.

3.In Vivo Mechanistic Data: The in vivo data convincingly show that Nec-1 treatment ameliorates disease. However, the mechanism proposed from the in vitro work (Nrf2/HO-1 suppression → ROS → necroptosis) is not thoroughly validated in the mouse model.

  •  

4.The introduction is comprehensive but could be slightly more focused. The connection between ROS, Nrf2, and necroptosis could be introduced more explicitly earlier on.

Round 2

Reviewer 3 Report

Comments and Suggestions for Authors
  1. Figure 1B, it is very strange data where CVB3 is 4 times higher than control ?
  2. CVB3 3D suddenly expression level high in 9h where 6h and 3h almost no signal . Why ?
  3. Every experiment we need proper control. In you data, for example PI staining where is positive control ?
  4. Need quantification for figure 2F
  5. Figure 5 we need p-Nrf2 and Keap-1. Without this data we cannot make any conclusion. 
  6. For RIP 1 actually 6h and 9h have not significant different. how can you explain ?
  7. How about Masson's trichrome stain data ?
  8. Fig 6 mechanism are not clear. Please explain briefly  
  9. Fig 3C internal control is not good. so it is hard to tell what really alter. 
  10. Fig 5 A no molecular weight ?
  11. Could you provide the all quantification WB data ? Please provide us in excel sheet. We may need to check .
  12. What is the method for WB data quantification ?
  13. Why plagiarism too high ?
  14. It is very important to know how to show supplementary data ? for example for WB triplicate data, you should present in MS power point. All blot should be in MS power point file with kDa. Must be triplicate. 
  15. All immunofluorescence microscopy data should show high and low magnify images. 
